# Predicting the presence of inline citations in academic text using binary classification

**Peter Vajdečka**
Prague University of
Economics and Business
Prague, Czechia
vajp02@vse.cz

**Elena Callegari**
University of Iceland
Reykjavík, Iceland
ecallegari@hi.is

**Desara Xhura**
SageWrite ehf.
Reykjavík, Iceland
dxhura@gmail.com

**Atli Snær Ásmundsson**
University of Iceland
Reykjavík,
Iceland
atliasmunds@gmail.com

## Abstract

Properly citing sources is a crucial component of any good-quality academic paper. The goal of this study was to determine what kind of accuracy we could reach in predicting whether or not a sentence should contain an inline citation using a simple binary classification model. To that end, we fine-tuned SciBERT on both an imbalanced and a balanced dataset containing sentences with and without inline citations. We achieved an overall accuracy of over 0.92, suggesting that language patterns alone could be used to predict where inline citations should appear.

## 1 Introduction

Providing accurate, relevant citations is an essential part of academic writing. Not only do citations allow authors to better contextualize the results of the paper, but they also lend credibility and authority to the claims made in the article. Failing to give credit to existing research when credit is due, on the other hand, is taken to show a lack of academic integrity, and is strongly frowned upon by the academic community. Appropriately adding citations, however, is not trivial: even humans sometimes struggle to determine where inline citations should go, and what should or should not be cited. This is particularly true in the case of junior academics and students (Vardi, 2012) (Carson et al., 1992) (Pennycook, 1996). In the context of *automatic* text evaluation, determining where citations should go is even less straightforward. One way in which one could automatically determine whether a given paragraph requires (additional) inline citations is through automatic plagiarism detection systems. However, processing a document to determine whether some sections of it have been plagiarized can require a considerable amount of time, particularly if the document exceeds a certain length. Building a plagiarism checker is also complicated, as the process requires scanning the full web for documents, and possibly obtaining access to research articles that might lay behind a paywall. Finally, results might not always be accurate (Kohl Kerstin, 2012), as the checker might fail in finding similarities between concepts simply because sentences that are identical in meaning have been expressed through a different formulation. Because of these downsides, we were interested in exploring how much mileage we could get out of a simple binary classification experiment trying to predict whether or not a given sentence should include an inline citation. In particular, we reasoned that it should be possible to predict at least to some extent whether a sentence should contain an inline citation simply by looking at the presence vs. absence of specific lexical cues. For example, verbs such as "claimed", nouns such as "authors" and phrases such as "as seen in" tend to appear together or in the vicinity of inline citations. The same holds true of some capitalized nouns (e.g. "Attention", "Minimalism").

### 1.1 Related Work

References play an essential role in academia and as such have been the focus of several NLP studies (Iqbal et al., 2021). Some of the properties that researchers have traditionally focused on are extracting the polarity of inline citations (is the referenced article negatively or positively mentioned?) (Abu-Jbara et al., 2013), and determining the purpose of inline citations (Viswanathan et al., 2021). Our paper builds on a body of research that has attempted to predict the "citation worthiness" (Cohan et al., 2019) of sentences, i.e. whether or not a given sentence should contain an inline citation. Several approaches have been suggested to determine the citation worthiness of text, see in particular (Beel et al., 2016), (Färber and Jatowt,

2020) and (Ma et al., 2020) for an overview. We have also seen an increased tendency towards using references as a way to build knowledge graphs (Viswanathan et al., 2021) and speed up the search for relevant research articles. There is also a tendency towards using references to aid automated text summarization (Yasunaga et al., 2019).

## 1.2 Motivation

Developing shallow automated techniques that can detect whether or not a sentence should contain an inline citation has several practical applications. A shallow inline-citation predictor can be used to (i) help academics identify forgotten inline citations, i.e. citations that the author meant to add at the review stage but ultimately forgot to include, (ii) guide junior researchers in the paper-writing process, flagging concepts or ideas that might require attribution, (iii) improving the coverage of automatic essay analyzers, and (iv) in the context of natural language generation, decreasing the chances of committing plagiarism by flagging passages that might require a citation.

## 2 Preparing the Data

To determine what types of inline citation styles are used in different research disciplines, we randomly selected two articles for each of the following 18 research fields: Medicine, Biology, Chemistry, Engineering, Computer Science, Physics, Math, Psychology, Economics, Political Science, Business, Geology, Sociology, Geography, Environmental Science, Art, History, Philosophy. After analyzing these 36 articles, we concluded that most of the articles adopted the IEEE, APA or the Chicago reference styles.

We first created an initial dataset consisting of 2000 research articles; these were randomly selected from the ArXiv and PubMed datasets (Cohan et al., 2018) that are freely available on the Huggingface Datasets library (Lhoest et al., 2021) (`https://huggingface.co/datasets/scientific_papers`).

These 2000 articles were subsequently processed to discard articles with a citation pattern other than the IEEE, APA or Chicago reference styles. The pre-processing task of detecting inline citations was handled through a simple Python script. Using regular expressions, different kinds of citation styles were mapped to corresponding regex capture patterns. We started by writing regexes that would match the three citation styles that we identified as the most frequently used: IEEE, APA and Chicago. Later on, we also decided to include the alpha BibTeX style, as that appears to be quite frequently used in ArXiV papers. The Python script did the following: first, every given citation pattern was extracted from the article's plain text. Then, the style with the highest capture count was set as the article's default style. This means that even when the extraction process found inline citations that matched a style that was not the article's primary citation style, the script was still able to identify the primary style. Finally, the inline citations matching the primary style were substituted with an -ADD-CITATION- token; this step is important as it allowed us to generalize across different referencing styles. If for some reason no citation style was detected, the token replacement failed, and the article was discarded from further analysis.

We then created a second dataset by taking all the articles with IEEE, APA or Chicago as reference styles and by (i) breaking down the original text into sentences, and assigning each sentence to a separate entry, (ii) assigning different labels to entries containing inline citations and entries not containing inline citations, and (iii) removing the -ADD-CITATION- token throughout the dataset. This second dataset features 411'992 sentences (entries), of which 54'735 contain an inline citation (see Table 1). The dataset is accessible at `https://github.com/elenaSage/InlineCitationSet` and is free to use. This second dataset is the dataset we used for the classification experiments that we describe below.

| No Citation | Contains Citation | Total |
|---|---|---|
| 357257 | 54735 | 411992 |

Table 1: Composition of Inline Citation Dataset

## 3 Classification model

In our research, we intend to train a classification model that can determine whether a sentence should contain a citation (positive class) or not (negative class) depending on the text input. In the first column of the Table 2, an example of the input text is displayed. The model we aim to train for this input text should predict that a citation must be present; this is a positive class prediction. If the

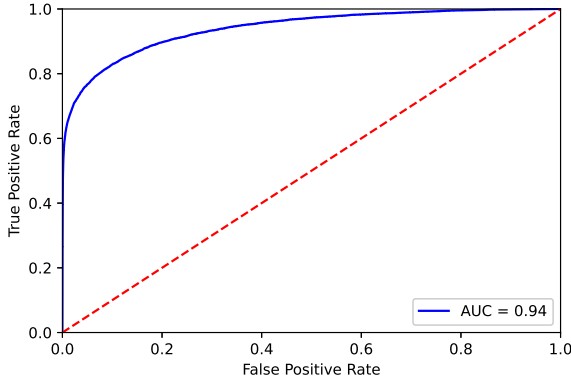

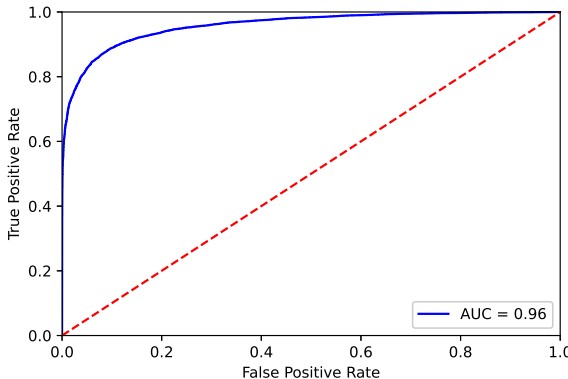

Figure 1: ROC curve on testing imbalanced dataset

Figure 2: ROC curve on testing balanced dataset

model predicts a negative class, it would mean that the text should not contain any citation.

In recent years, BERT-based language models (Devlin et al., 2019) have achieved state-of-the-art performance in numerous NLP classification tasks. Due to their pre-training on massive corpora and fine-tuning for a specific downstream purpose, these models can acquire accurate language representations.

Our Inline Citation dataset includes scientific data containing science-specific terminology. Because of that, we decided to encode texts for the classification task using the BERT architecture that has been pre-trained on scientific texts, i.e. the SciBERT model (Beltagy et al., 2019). Exactly like BERT, SciBERT contains 30K word-piece tokens, but unlike BERT its vocabulary is pertinent to the scientific area. In the scientific domain, SciBERT outperforms BERT in a variety of tasks (Beltagy et al., 2019) and achieves SOTA performance in multi-class text classification on the SciCite dataset (Cohan et al., 2019). It has been demonstrated that fine-tuned uncased SciBERT with SciVocab followed by a linear layer produces the best results for scientific data (Beltagy et al., 2019) or for citation context classification (Maheshwari et al., 2021). Therefore we use this model in each experiment.

## 4 Fine-tuning SciBERT

Research papers generally contain more sentences without inline citations than sentences with citations, which leads to having more examples for the "no citation" class. Performing classification tasks using imbalanced datasets poses multi-

ple challenges, the most prominent being the bias towards the most represented class (He and Garcia, 2009). There are multiple studies that try to counteract this phenomenon by bringing more balance in the distribution of classes within the same dataset (see for example (Mohammed et al., 2020) or (Krawczyk, 2016)). Two well-known techniques in direction of balancing are undersampling and oversampling. Undersampling however also presents drawbacks, the most important one being the loss of information that might be captured by the most represented class. With this in mind, we decided to run classification experiments on both the full (imbalanced) dataset and a more balanced subset of the dataset which we obtained by undersampling the data. We divided both the balanced and the imbalanced dataset into a training subset (60%), a validation subset (20%) and a test subset (20%), resulting in a "60:20:20" split. The split was then modified so that the proportion of positive (sentences containing a citation) to negative (sentences not containing a citation) texts in each subset would not be altered following the split (see Table 3).

Next, we fine-tuned all SciBERT parameters end-to-end utilizing the training and validation subsets. For fine-tuning, we adhered primarily to the similar design and optimization decisions utilized in articles (Beltagy et al., 2019; Devlin et al., 2019). We used the ReLu activation function in linear one-layer feed-forward classifier which inputs the last hidden state of the [CLS] token. In other words, this last hidden state of the [CLS] token is utilized as the sequence's features to feed the classifier.

We experimented with numerous hyper-

Table 2: An illustration of text input and prediction output

| Input sentence | Class |
|---|---|
| In particular, our colored pebbles generalize and strengthen the previous results of Lee and Streinu and give a new proof of the Tutte-Nash-Williams characteri- zation of arboricity. | Positive |
| The tidal friction theories explain that the present rate of tidal dissipation is anomalously high because the tidal force is close to a resonance in the response function of ocean. | Positive |
| A k-map-graph is a graph that admits a decomposition into k edge-disjoint map-graphs. | Negative |

| Dataset type | Class | Training subset | Validation subset | Testing subset |
|---|---|---|---|---|
| Balanced | Contains citation | 32831 | 10957 | 10947 |
| | No citation | 36085 | 12015 | 12025 |
| Imbalanced | Contains citation | 32739 | 11049 | 10947 |
| | No citation | 214455 | 71350 | 71452 |

Table 3: Dataset split

parameters for fine-tuning with both datasets. We fine-tuned for 2 to 5 epochs using batch size 16, 32 or 50 and learning rate of 5e-5, 5e-6, 1e-5 or 2e-5, with a dropout of 0.1 or without dropout. We optimized cross-entropy loss with the assistance of the AdamW optimizer (Kingma and Ba, 2014). The best results were obtained when the models were fine-tuned for 2 epochs with a batch size of 50 samples and a learning rate of 5e-5 without dropout, followed by a linear warmup and linear decay (Devlin et al., 2019); this was the case for both the balanced and the imbalanced dataset. We used softmax to determine probabilities for predictions, with a threshold of 0.7 proving optimal, meaning that sentences with a calculated probability greater than 0.7 are predicted to be positive, i.e. they are predicted to contain an inline citation.

## 5 Discussion

In our work, we always consider positive labels as a class of those input texts that contain an inline citation. This means that we always understand True-Positives (TP) as correctly predicted texts that contain an inline citation (see graph 1 and graph 2). This is also analogous to the Precision and Recall calculations and the derived F-score in graphs 3 and 4, and the metrics in Table 4 below. The focus is mainly on this class of inline citations as positive, since it is definitely a minority with respect to quantity, which makes the

problem more challenging.

We report the results of our two experiments in Table 4. We see that balancing the dataset by undersampling helped to significantly reduce the bias towards the most represented class, increasing the recall of the least represented class (=sentences containing an inline citation) from 0.63 to 0.84.

Since we used both balanced and imbalanced datasets, useful performance indicators include the Area under the Curve AUC for the precision-recall curve PR or the Receiver Operating Characteristic curve ROC (Bradley, 1997; Hanley and McNeil, 1982). Figure 1 and figure 2 reveal that the ROC curves are nearly comparable in both datasets, with the imbalanced dataset having a slightly lower AUC value of 0.94 against that of 0.96 for the balanced dataset. For imbalanced data, however, a PR plot is advised (Sun et al., 2009; Gu et al., 2009); our PR plots are depicted in figure 3 and 4. The imbalanced dataset's PR curve follows a different path than the balanced dataset's PR curve, which is also reflected in its considerably lower AUC value (=0.84) compared to that of the balanced dataset (=0.96).

## 6 Conclusion

The goal of this paper was to determine how effective binary classification models can be at predicting whether or not sentences appearing in academic articles should contain an inline citation.

**Citation prediction**

| Approach | Precision | Recall | F1 score | Accuracy |
|---|---|---|---|---|
| Balanced SciBERT validation | 0.93 | 0.84 | 0.89 | 0.90 |
| Balanced SciBERT testing | 0.93 | 0.84 | 0.88 | 0.89 |
| Imbalanced SciBERT validation | 0.92 | 0.63 | 0.75 | 0.94 |
| Imbalanced SciBERT testing | 0.92 | 0.64 | 0.75 | 0.94 |

Table 4: Prediction results

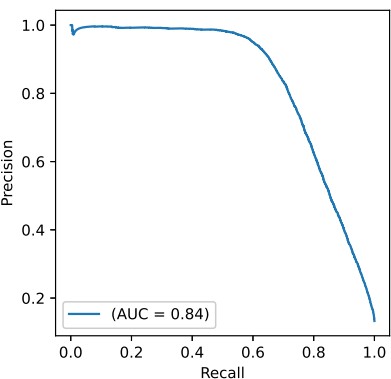

Figure 3: PR curve on testing imbalanced dataset

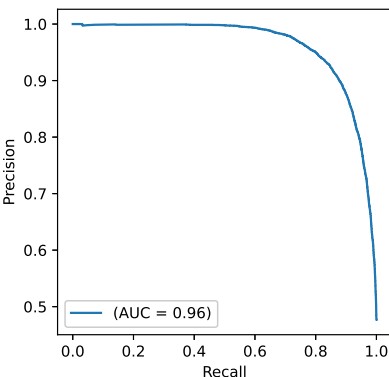

Figure 4: PR curve on testing balanced dataset

To that end, we used regular expressions to identify inline citations in published research papers, and then created a dataset composed of 411k sentences, where approximately 54k contained inline citations. We then ran a fine-tuned SciBERT classifier on both a balanced and imbalanced dataset, achieving an overall accuracy of over 0.92. This result shows that language patterns alone could be used to predict the presence of inline citations in academic text with a reasonable degree of accuracy. We presented the problem as a binary clas-

sification task on the sentence level, i.e. we only considered the target sentence and did not consider the context in which the sentence appeared, for example by also looking at the sentences appearing before and after the target sentence. Taking into account the previous and the following sentence could be worthwhile in that some inline citations scope over multiple contiguous sentences rather than just refer to a single sentence (i.e. the concept of "citing area" first mentioned in (Nanba and Okumura, 1999)). The sentences contained in the Inline Citation Dataset however are all sequential: they come in the same sequence as they were found in the original paper. This means that information on the context in which a given target sentence appears is already available in our dataset. This paves the path for further experiments that take contextual sentential information into account, such as using transformers to predict in which position inline citations should appear.

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
