# OpenReview forum: "Predicting the presence of inline citations in academic text using binary classification"
_NoDaLiDa/2023/Conference — NoDaLiDa 2023_

### Official Review · Reviewer_vb4X · 2023-03-06
**Interesting novel task**

**Rating:** 6
**Confidence:** 4

**Review:**

The paper describes an interesting novel: predict if a sentence should contain a citation. Altogether, the paper is quite well written but a lacks a bit of context and relation to previous work. Also, the motivation, usecase and main findings could be explained more clearly.

I propose to do so already in the motivation. The introduction does clearly mention (lines 96ff) what the use cases are, but such motivation could also make the abstract more relevant.

Despite the fact that this is a short paper, I propose to include a short related work section. There is quite some work on citation type detection which is clearly related (see N19-1361 as a starting point for literature research); also the area of citation polarity detection could be mentioned (see N13-1067).

The experimental setup is not sufficiently clear. What is the unit of classification? Is it sentences? Could Table 1 also include that mention and perhaps also state from which domain the sentences come? This table is currently a bit boring with this single row and this information would be valuable.

Further, I propose to start Section 3 with a clear formal statement of the classification task (what's the input, context, output).

I appreciate the depiction of ROC curves, but what is not clear here is the Precision/Recall is counted across all classes as an average or if only one class is shown. Also, showing F/R/P would be a more standard way of evaluating text classification models.

Please state clearly if the data that you collected is available.

Finally, I propose to increase the style of the paper: Don't change the font size (particularly not the aspect ratio!) of fonts in tables. Move implementation details into an appendix and focus in the main paper on the research; not on engineering steps. Do not cite arxiv papers if a published version is available


**Paper Type:**

Short paper

---

### Official Review · Reviewer_7FTb · 2023-03-09
**The tool described in the paper is useful and could be beneficial for many.**

**Rating:** 7
**Confidence:** 4

**Review:**

The paper is written in a clear and consistent way. The choice of the approach and the steps of the workflow are well argued. The tool described in the paper is useful and could be beneficial for many.
On the other hand, it was hard to understand if and how many experts gave feedback. Was there something improved in the texts?
It would be good to have some examples of an excellent and some of faulty results or errors.



**Paper Type:**

Short paper

---

### Decision · Program_Chairs · 2023-03-17

Accept